# Epigenetic assessments of alcohol consumption predict mortality in smokers at risk for lung cancer in the prostate, lung, colorectal and ovarian cancer screening trial

Robert Philibert[1,2]*, Steven R.H. Beach[3], James A. Mills[1], Kelsey Dawes[2], Richard M. Hoffman[4], Jessica C. Sieren[5], Ellyse M. Froehlich[1], Kaitlyn M. deBlois[1], Jeffrey D. Long[1,6]

1 Department of Psychiatry, University of Iowa, Iowa City, Iowa, United States of America, 2 Behavioral Diagnostics LLC, Coralville, Iowa, United States of America, 3 Center for Family Research/Department of Psychology, University of Georgia, Athens, Georgia, United States of America, 4 Division of General Medicine, Department of Internal Medicine, University of Iowa, Iowa City, Iowa, United States of America, 5 Department of Radiology, University of Iowa, Iowa City, Iowa, United States of America, 6 Department of Biostatistics, University of Iowa, Iowa City, Iowa, United States of America

* robert-philibert@uiowa.edu

## Abstract

DNA methylation at cg05575921, an established biomarker for smoking predicts risk for lung cancer (LC). Although heavy alcohol consumption (HAC) frequently accompanies smoking, the relationship of HAC to overall mortality in those at risk for LC is not well known. Determining the contribution of HAC to mortality in those who smoke is important because HAC is also a major driver of mortality and is potentially treatable. To help answer this question, we examined the relationship of epigenetic biomarkers of smoking (cg05575921) and chronic heavy alcohol consumption (Alcohol T Score, ATS) in a cohort of 92 LC cases and 402 age, sex, ethnicity and smoking history matched controls from the Prostate, Lung, Colorectal and Ovarian (PLCO) Screening Trial to all-cause mortality using proportional hazards survival analysis. We found that ATS values significantly predicted risk for all-cause mortality in those smokers who developed (p < 0.03) and did not develop lung cancer (p < 0.0001). When mortality data were analyzed using median splits, those who did and did not incur lung cancer with ATS values <3.6 lived 5.6 years and 3.2 years more, respectfully, than those with ATS values >3.6. Interestingly, in this group of 494 smokers or former smokers, after adjusting for the occurrence of lung cancer, cg05575921 methylation did not predict mortality. In summary, we found that excessive alcohol consumption is a significant risk factor for all-cause mortality in those at risk for LC and suggest that lung cancer screening efforts to address problem drinking could increase survival.

**Data availability statement:** We have uploaded the minimal dataset to the Open Science Foundation. The link is https://osf.io/4fhj8/.

**Funding:** National Institutes of Health 1R44CA285136.

**Competing interests:** Dr. Philibert is the Chief Executive Officer of Behavioral Diagnostics. The use of cg05575921 to assess smoking status is covered by existing and pending patents including US Patents 8,637,652 and 9,273,358, while the use of DNA methylation to assess alcohol and predict is covered by existing patents and pending patent claims including European Union Patent 3149206. On behalf of Drs. Philibert, Behavioral Diagnostics and the University of Iowa have a filed an intellectual property claim on the use of DNA methylation to predict AWS and related phenomena.

**Abbreviations:** ACS, acute coronary syndrome; ATS, alcohol T score; CDC, centers for disease control; CDT, carbohydrate deficient transferrin; CHD, coronary heart disease; HAC, heavy alcohol consumption; LC, lung cancer; MSdPCR, methylation sensitive digital polymerase chain reaction; PLCO, Prostate, Lung, Colorectal and Ovarian screening trail.

## Introduction

Lung cancer (LC) is the leading cause of cancer death in the United States [1]. Epidemiological studies have suggested that smoking accounts for 90% of lung cancer mortality [2,3]. This effect of smoking is dose-dependent, and studies have shown that cg05575921 methylation, an established biomarker of smoking, predicts likelihood of LC [4,5].

Those who smoke heavily often also drink alcohol heavily [6,7]. However, the impact of excessive alcohol consumption in those who smoke is not well described. In part, this lack of understanding is secondary to the difficultly of separating the deleterious effects of smoking from that of drinking.

Further advances in epigenetics may help resolve these issues. Using the same approach used to identify the cg05575921 methylation site in 2012, we developed a metric of chronic HAC, defined as consuming ≥ 6 drinks per day for 8 or more weeks, called the Alcohol T Score (ATS) [8]. The ATS uses methylation sensitive digital polymerase chain reaction (MSdPCR) to quantify methylation at four CpG sites that are sensitive to alcohol but unaffected by smoking [9]. In abstinent individuals, ATS is zero-centered metric with a standard deviation of 2.2 that non-linearly increases as a function of increasing chronic alcohol consumption [10]. The performance of the ATS has been examined in ten studies (for review see [10]), including three direct comparisons against carbohydrate deficient transferrin (CDT), an established biomarker of heavy drinking [11]. In these three studies, the ATS outperformed the CDT in predicting chronic HAC, alcohol withdrawal and alcohol related immune cell changes [9,12,13].

In at least one disease associated with HAC, coronary heart disease [14], the ATS predicts mortality. Building on work showing that the CpG sites surveyed in the ATS predict mortality in large population cohorts, we examined the relationship of ATS values to survival in subjects admitted for acute coronary syndrome. We found the ATS strongly predicted survival, even more so than age or degree of coronary artery obstruction.

Using self-report data of smoking and drinking, epidemiological studies have shown that smoking is also associated with HAC [6,7]. This is consistent with our epigenetic studies which show strong correlations between cg05575921 methylation and ATS values. However, whether ATS values predict survival independently of the effects of smoking in those with a history of smoking is not well characterized. Because the most recent recommendations by the United States Preventative Services Task Force has recommended reducing the lower limit for low dose cancer risk screening from 30 pack years to 20 pack years of smoking and there has been an increased emphasis by the Surgeon General on the need for alcohol prevention and treatment, the possibility that lung cancer risk screening visits could be expanded to include thorough assessments of alcohol consumption could represent an opportunity for clinicians to further reduce substance use related morbidity and mortality.

In this pilot study, we examine the relationship of smoking and drinking intensities to survival using these two epigenetic metrics and the bioresources of lung cancer cases and matched controls from the Prostate Lung, Colorectal and Ovarian (PLCO) Cancer Screening Trial [15].

## Methods

### Study design

The design and methods of the PLCO trial have been previously reported [15]. In brief, the PLCO Cancer Screening Trial included approximately 148,000 individuals enrolled between 1993 and 2006 who were randomized to either an intervention or usual care arm at 10 screening centers in the United States, who were then followed for up to 13 years with respect to key clinical outcomes.

All subjects in the study provided written informed consent. The overall project was approved by the institutional review board at the National Cancer Institute and at each of the ten recruitment sites. The de-identified clinical data for the current study were provided by Etiology and Early Marker Studies (EEMS) coordinating center of the Cancer Data Access System. The DNA specimens for this study were obtained from National Cancer Institute's Cancer Genomic Research Laboratory (Frederick, MD).

### Ethics statement

As this study used only de-identified data and samples provided by the repository, it was exempt from human subjects review.

### Study sample

The DNA samples and data for the 494 subjects, included in this study are a subset of a larger study of 4910 subjects with whom the goal is to construct an algorithm incorporating cg05575921 methylation to predict lung cancer. The strategy employed by EEMS to select the overall sample cohort matched each case to 3 controls with respect to gender, race, age at randomization, smoking history and year of randomization by the EEMS coordinating center staff. The 494 subjects included in this study represent every smoker or former smoker from the first six DNA plates provided by Cancer Genomic Research Laboratory and -for whom ATS values were also determined as part of routine quality control measures.

### Cg05575921 determinations

500 ng samples of DNA from each subject were bisulfite converted using a Qiagen (Germany) Epitect kits. Then for cg05575921 methylation, a 3 µl aliquot of each sample was pre-amped, diluted 1:3000, and then PCR amplified using fluorescent, dual labeled primer probe sets specific for the locus (Behavioral Diagnostics, Coralville, IA), in combination with digital PCR reagents and a QuantStudio Absolute Q Digital PCR System from ThermoFisher (Hercules, CA). Then, methylation values ((C/C + T) ratios) were determined using the proprietary software.

### ATS determinations

ATS values for each subject were determined using fluorescent, dual labeled primer probe sets from Behavioral Diagnostics specific for the four loci included in the ATS, cg02583484, cg04987734, cg09935388 and cg04583842, and droplet digital PCR machinery and reagents from Bio-Rad (Carlsbad, CA) as previously described [9,10]. For each locus, a 3 µl sample of bisulfite converted DNA was pre-amped, diluted 1:1500, and then amplified. Then methylation for each sample was determined using a QX-200 droplet reader and proprietary Bio-Rad software. Finally, the Z-scores for each CpG site was produced by subtracting the previously established mean of methylation at the locus in abstinent controls, then dividing the result by the standard deviation of the controls at that locus, then summing the four scores to form the ATS.

### Data analysis

Initial analysis of data was conducted using JMP Version 17 (SAS Institute, Cary, SC). Group comparisons for both normally and non-normally distributed continuous variables were conducted using t-tests, and Wilcoxon rank sums,

respectively. The event of interest was all-cause mortality, and the time metric was days on study (study entry was time 0). Kaplan-Meier curves were created for controls and LC cases using the median split of ATS values to illustrate the impact of alcohol consumption on survival for each group. Survival was more formally assessed using Cox proportional hazards models, with separate models fit for controls (N = 402) and LC cases (N = 92). To examine the interrelationship of smoking and drinking intensity, models included cg05575921 methylation and ATS values, adjusting for mortality risk factors age and sex. Another measure of smoking intensity, pack years, was also considered in the model. Plausibility of the proportional hazards assumption was assessed using graphical displays and appeared to be reasonable.

## Results

Table 1 gives the key demographic characteristics for the 494 subjects (all were smokers or former smokers, controls did not have lung cancer, whereas cases did). Subjects were largely White and in their early 60s. As would be expected by the matching paradigm instituted by EEMS, there were minimal differences between the cases and controls with respect to age, sex, and race. However, the number of cases placed on the first six plates supplied by CGR (n = 92) was lower than would be expected in a 3:1 matching paradigm.

**Table 1. Key Demographic and Clinical Characteristics of the Subjects.**

|  | Controls (No Cancer) | | Cases (Lung Cancer) | |
|---|---|---|---|---|
|  | Male | Female | Male | Female |
|  | N = 255 | N = 147 | N = 58 | N = 34 |
| Age (years) mean (SD) | 62.4 ± 4.8 | 63.3 ± 5.6 | 62.6 ± 4.7 | 62.4 ± 4.9 |
| Race: |  |  |  |  |
| White, non-Hispanic | 223 | 135 | 48 | 31 |
| Black, non-Hispanic | 11 | 8 | 6 | 2 |
| Hispanic | 5 | 2 | 1 | – |
| Asian | 14 | – | 3 | 1 |
| Pacific Islander | 1 | 1 | – | – |
| American Indian | 1 | 1 | – | – |
| Ethnicity: |  |  |  |  |
| Hispanic | 6 | 2 | 1 | – |
| No | 240 | 144 | 55 | 34 |
| Missing | 9 | 1 | 2 | – |
| Pack Years Smoked: mean (SD) | 43.6 ± 32.7 | 34.3 ± 25.6 | 54.1 ± 28.7 | 48.7 ± 42.1 |
| Current Smoker: N (%) |  |  |  |  |
| Yes | 100 (39%) | 57 (39%) | 25 (43%) | 16 (47%) |
| No | 155 (61%) | 90 (61%) | 33 (57%) | 18 (53%) |
| Methylation |  |  |  |  |
| ATS (unitless) | 3.9 ± 3.3* | 3.2 ± 3.1 | 4.3 ± 3.8 | 4.7 ± 3.1 |
| Cg05575921%: mean (SD) | 61.6 ± 21.3 | 64.5 ± 19.5 | 54.2 ± 20.6 | 58.3 ± 20.5 |
| Mortality during follow up: N (%) |  |  |  |  |
| Overall1 | 23 (48%)* | 56 (38%) | 48 (83%) | 27 (79%) |
| Lung cancer | -- | -- | 35 (60%) | 18 (53%) |
| Survival (days) | 6172 ± 1927* | 6581 ± 1808 | 5914 ± 1875 | 5327 ± 2154 |

SD = standard deviation.

*Different than paired female value at p < 0.05.

LC cases were not more likely than controls to be current smokers (45% vs 39%, Chi-square p<0.34). Male (54±28 vs 44±33, p<0.03) and female (49±42 vs 34±26, p<0.02) LC cases had greater pack year histories than their sex matched counterparts.

Cg05575921 are non-normally distributed in both the case and control subjects (See Fig 1) with the modal peak occurring between 80–82% for controls but at 72–74% for case subjects. Male case subjects (54.2±20.6 vs 61.6±21.3, Wilcoxon p<0.006) had lower cg05575921 methylation values (i.e., greater smoking intensity) than their sex matched controls. Female case subjects had non-significantly lower cg05575921 levels than their matched controls (58.3±20.5 vs 64.5±19.5, Wilcoxon p<0.07).

ATS values were slightly right skewed in both case and control subjects (see Fig 2). Female case subjects (4.7±3.1 vs 3.2±3.1, p<0.02, Wilcoxon), but not male case subjects (4.3±3.8 vs 3.9±3.3, NS) had higher ATS values (i.e., greater alcohol intake) than their matched controls.

Fig 3 illustrates the relationship between ATS, which increases as a function of drinking intensity, and cg05575921, which decreases as a function of smoking intensity, values. Overall, the values were strongly negatively correlated, and a linear fit explaining nearly half of the variance (Bivariate Fit, Adjusted $R^2$=0.45, p<0.0001).

The subjects were followed for a median of 20.7 years (interquartile range; 15.0–23.3 years). LC cases were much more likely to die (81.5% vs. 44.5%, p<0.0001, Chi-Square) than control subjects during the follow up period (Table 1). Death certificates listed LC as the primary cause of death for 35 male LC cases and 18 female LC cases.

Table 2 presents the parameter estimates for the multivariable Cox models. Age was significantly positively associated with mortality risk and males had higher risk than females. In addition, ATS values were significantly positively associated with increased mortality risk for controls (p<0.0001; HR=1.14 for a 1-unit increase in ATS) and for LC cases (p<0.03; HR=1.10 for a 1-unit increase in ATS). Cg05575921 methylation was not significantly associated with mortality risk for either cases or controls in this analysis. In separate modeling not presented, models stratified by sex yielded similar parameter estimates for ATS and cg05575921 and the addition of pack year smoking history was not associated with mortality risk for either cases or controls in this analysis. Similarly, because the interaction term between the ATS and cg05575921 was not significant, it also was not included in the final model.

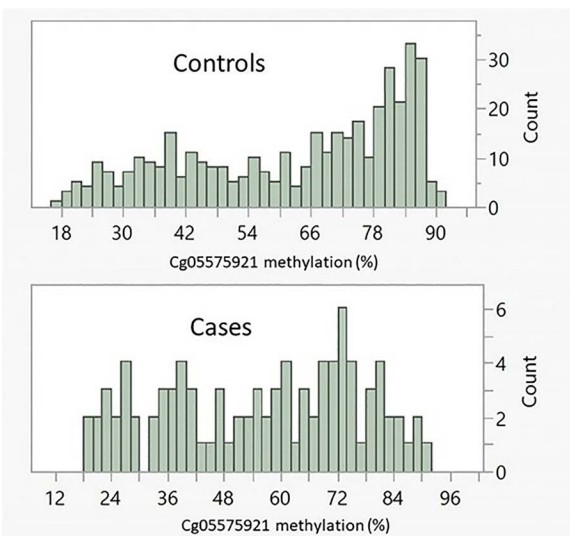

**Fig 1. The distribution of cg05575921 values in controls (n =402) and LC case (n=92) subjects.**

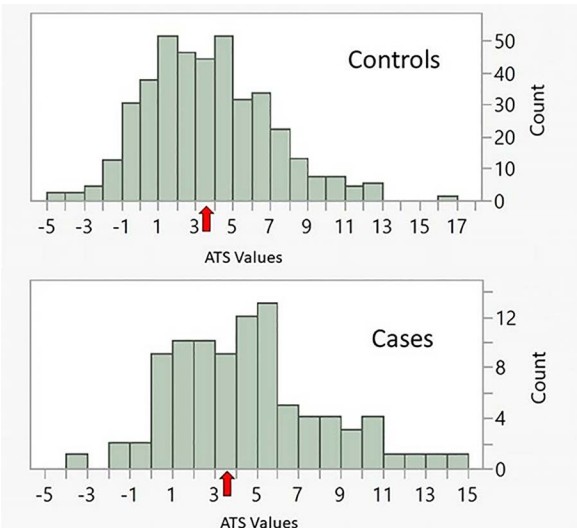

**Fig 2. The distribution of ATS values in values in controls (n = 402) and LC case (n = 92) subjects.** The red arrows indicate the cutoff of 3.5 for HAC as shown in Miller et al., 2019.

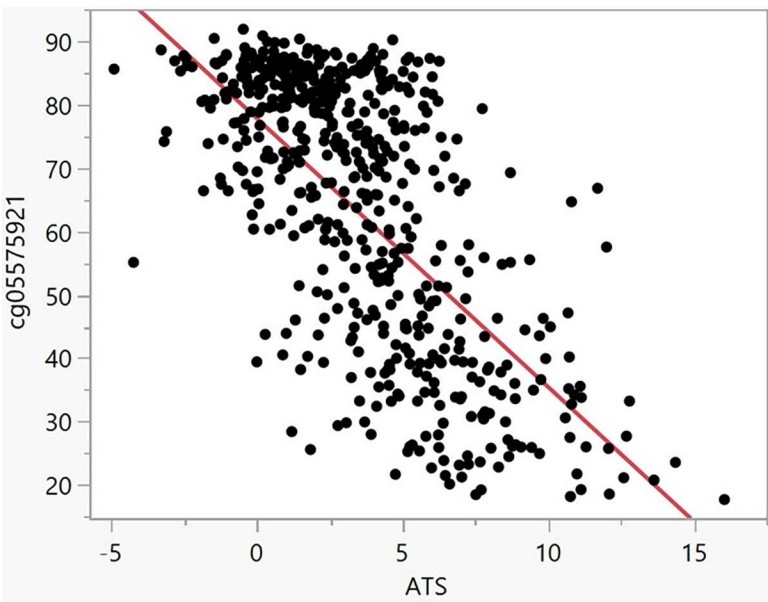

**Fig 3. The relationship of cg05575921 to ATS values in all 494 subjects (Adjusted $R^2$ = 0.45, p < 0.0001).**

Fig 4 graphically illustrates the effects of alcohol consumption on survival in the two groups using median splits with respect to ATS values. Controls whose ATS was < 3.6 had an estimated median survival of 9226 days while those with ATS ≥ 3.6 had a median survival of 8067 days (a 1159-day or 3.2 year difference). The median survival for those LC cases with ATS < 3.6 was 7389 days while those with ATS ≥ 3.6 had a median survival of 5362 days (a 2027-day or 5.6 year difference).

**Table 2. Parameter Estimates for Proportional Hazards Modeling of Survival.**

| Variable | Controls (No Lung Cancer, n=402) | | | Cases (Lung Cancer, n=92) | | |
|---|---|---|---|---|---|---|
| | Parameter Estimate | Standard Error | p-value | Parameter Estimate | Standard Error | p-value |
| Age | 0.12 | 0.016 | **<0.0001** | 0.050 | 0.025 | **<0.05** |
| Sex (M) | 0.41 | 0.16 | **<0.02** | 0.29 | 0.25 | <0.25 |
| Cg05575921 | -0.0037 | 0.0050 | <0.46 | -0.011 | 0.0080 | <0.17 |
| ATS | 0.13 | 0.032 | **<0.0001** | 0.099 | 0.043 | **<0.03** |

Significant findings are bolded. M=male.

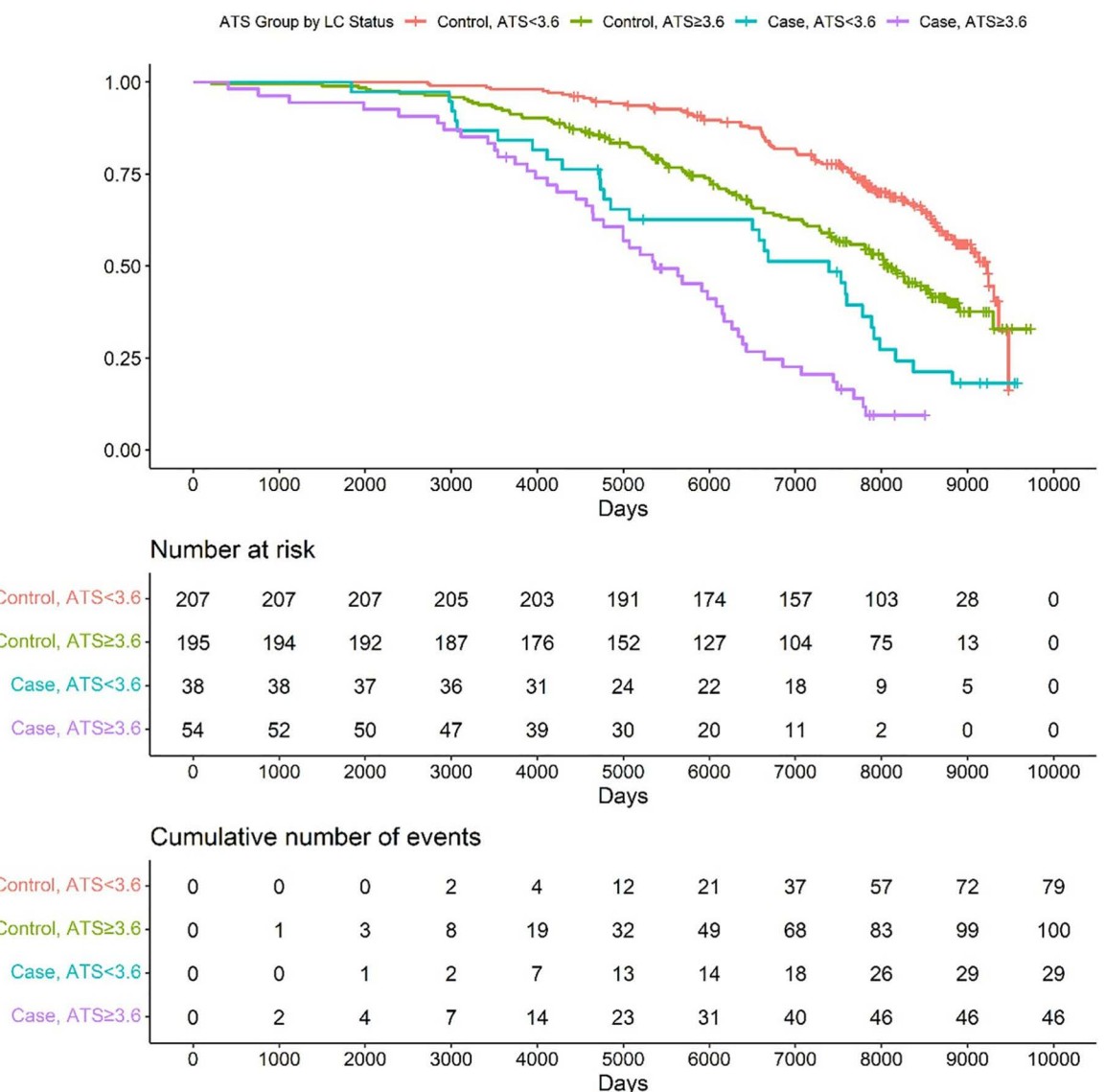

**Fig 4. Kaplan-Meier Curves by median split of the ATS values and LC status.** The four groups are: 1) LC cases with ATS<3.6, turquoise, 2) LC cases with ATS>3.6, purple, 3) Controls (no cancer) with ATS<3.6, red, and 4) Controls with ATS>3.6, green. The number of risk and cumulative number of events for each group are given below.

## Discussion

Using objective measures of substance use we showed that average alcohol consumption at the time of study intake significantly predicted of the overall mortality risk of subjects who were current or former smokers whether or not they were subsequently diagnosed with lung cancer. The fact that HAC makes a substantial impact on mortality in those who smoke is not surprising. Alcohol is thought to be the third leading cause of death in the United States by the Centers for Disease Control (CDC). Critically, these estimates of the effect of alcohol are based largely on self-report of alcohol use. However, both clinical and epidemiological self-reports of alcohol consumption can be unreliable [12,16,17]. In fact, as Nelson and colleagues have pointed out, over a 15-year period of time, the amount of alcohol sold in the United States according to the National Institute of Alcohol Abuse and Alcoholism surpassed the amount reported in CDC's Behavioral Risk Factor Surveillance System by at least a factor of three [17]. The parameter estimates for the ATS listed in Table 2 suggest that for equal smoking intensity, the 62-year-old male who is abstinent from alcohol and does not develop lung cancer will live about 4 years longer than the average 62-year-old male with an ATS of 3.9 who does not develop lung cancer.

DNA methylation studies alone do not provide the in-depth information that is needed to understand how the excessive alcohol consumption causes mortality. The ATS was designed to predict heavy chronic alcohol consumption (6 or more drinks per day for 8 or more weeks) [9]. Binge drinking and other forms of alcohol intake of lesser duration, which may confer somewhat different health risks than chronic heavy consumption, are not perfectly correlated with the ATS. For example, in a study using a marker for recent (past 3 weeks) heavy alcohol consumption, referred to a ZSCAN25, we found only a 0.56 correlation between ZSCAN25 and the ATS values for 125 subjects admitted to the hospital for possible alcohol withdrawal with the ZSCAN marker better predicting the consequences of sustained recent drinking than the ATS in that study [12]. Critically, in our studies using both the ATS and cg05575921 assessments, the ATS has been highly correlated with smoking [8,9,12,13,18–22]. Therefore, we believe that our findings our with respect to mortality reflect the entirety of lifestyle risk biology, including risks from poor diets and lack of exercise, that track with the heavily tobacco use and chronic alcohol consumption [23].

Unfortunately, defining the exact relationship of smoking and drinking to the entire risk biology associated with their epigenetic signal will be difficult. Not only do these behaviors sort cross-sectionally, but when one risk behavior, such as smoking remits, the other risk factors such as excessive drinking or poor dietary choices tend to remit as well [19,24]. Consequently, deriving an exact quantitative understanding of the mortality associated each of these individual clinical risk factors that is captured by the cg05575921 and the ATS will be difficult.

Conversely, understanding how the drinking and smoking related biology conveys the excess mortality at the molecular level is achievablebut will require integrating the epigenetic findings with more traditional molecular, cellular, and serological assessments of human samples. Certainly, the molecular and cellular effects of smoking are well delineated [25]. But how they interact, if at all, with the effects of alcohol, is not as well understood. By incorporating a potentially powerful continuous objective measure like the ATS into existing analyses of those who currently smoke and drink, it may be possible to define potential cellular targets more precisely for preventive interventions.

After controlling for LC status, we did not find that cg05575921 predicted mortality. Clearly, cg05575921 predicts the occurrence of LC and in part, that is why these analyses are controlled for by lung cancer status [4,5]. However, it may well be as we have shown in those admitted for ACS [22] or in the FHS [26], that the effects of alcohol measured through this method on non-LC outcomes are greater than those that can be shown using cg05575921 with respect to CHD, and that with greater sample size, the effects assessed by cg05575921 on these outcomes, such as CHD, will become significant as well. Still, given the plethora of conditions for which both alcohol and smoking are risk factors, the use of this quantitative epigenetic approach may help more exactly parse the amount of risk for each of these disorders that can more directly be attributed to smoking as opposed to that associated with drinking.

Limitations of the study include the relatively small size of the cohort and the absence of other objective markers of alcohol consumption. The relatively small sample size in this study precludes determining the shape of the alcohol consumption curve with respect to survival.

Our findings, which need to be replicated in larger and more diverse populations, do have potential implications for lung cancer screening efforts. According to the United States Preventative Services Task Force guidelines, approximately 18 million smokers are potentially eligible for low dose computerized tomography [27]. If our findings are replicated, they suggest that these patients should be rigorously screened for excessive alcohol consumption, and when indicated, referred for alcohol treatment. Because combined alcohol and tobacco cessation treatment may be more effective than smoking cessation alone, this may improve patient outcomes [28]. Unfortunately, our data are silent as to whether incorporating smoking and drinking biomarker testing into lung cancer screening risk calculators will improve overall patient outcomes. We note that when combined with contingency management approaches, the use of biomarker testing can lead to marked improvements in abstinence rates for both smoking and drinking [29,30]. However, some patients have concerns about even verbal screening for alcohol consumption [25], suggesting that the use of this type of technology needs to be discussed by the medical community.

## Acknowledgments

This work is dedicated to the memory of Dr. Meg Gerrard of the University of Connecticut whose work to prevent the effects of alcohol and smoking on risk for cancer have had a profound and lasting impact on the investigative team and the field in general.

## Author contributions

**Conceptualization:** Robert Philibert, Steven R.H. Beach, Richard M Hoffman.

**Data curation:** Kelsey Dawes, Kaitlyn M. deBlois.

**Formal analysis:** Robert Philibert, James A Mills, Jeffrey D Long.

**Funding acquisition:** Robert Philibert, James A Mills, Kelsey Dawes, Jessica C. Sieren, Jeffrey D Long.

**Investigation:** Kelsey Dawes, Ellyse M. Froehlich.

**Methodology:** Robert Philibert, James A Mills, Ellyse M. Froehlich, Kaitlyn M. deBlois.

**Project administration:** Robert Philibert.

**Visualization:** Jeffrey D Long.

**Writing – original draft:** Robert Philibert, Steven R.H. Beach, James A Mills, Richard M Hoffman, Jeffrey D Long.

**Writing – review & editing:** Robert Philibert, Steven R.H. Beach, James A Mills, Kelsey Dawes, Richard M Hoffman, Jessica C. Sieren, Ellyse M. Froehlich, Kaitlyn M. deBlois, Jeffrey D Long.

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
