## [Decision Letter · Decision Letter 0]

3 Feb 2025

PONE-D-25-03475Epigenetics Assessments of Alcohol Consumption Predict Mortality in Smokers at Risk for Lung Cancer in the Prostate, Lung, Colorectal and Ovarian Cancer Screening TrialPLOS ONE

Dear Dr. Philibert,

Thank you for submitting your manuscript to PLOS ONE. After careful consideration, we feel that it has merit but does not fully meet PLOS ONE’s publication criteria as it currently stands. Therefore, we invite you to submit a revised version of the manuscript that addresses the points raised during the review process.

Your manuscript has been reviewed by two experts in the field. Both reviewers have identified significant value in your paper and acknowledge its contribution to the field. However, they have also highlighted areas that require substantial revisions before the manuscript can be considered for publication. I encourage you to carefully address each comment and resubmit the manuscript for further consideration. 

We look forward to reviewing the revised version of your manuscript

We look forward to receiving your revised manuscript.

Kind regards,

Chunyu Liu, PhD

Academic Editor

PLOS ONE

Journal requirements:   When submitting your revision, we need you to address these additional requirements. 1. Please ensure that your manuscript meets PLOS ONE's style requirements, including those for file naming. The PLOS ONE style templates can be found at https://journals.plos.org/plosone/s/file?id=wjVg/PLOSOne_formatting_sample_main_body.pdf and https://journals.plos.org/plosone/s/file?id=ba62/PLOSOne_formatting_sample_title_authors_affiliations.pdf. 2. We note that the grant information you provided in the ‘Funding Information’ and ‘Financial Disclosure’ sections do not match.  When you resubmit, please ensure that you provide the correct grant numbers for the awards you received for your study in the ‘Funding Information’ section. 3. Thank you for stating the following financial disclosure:  [National Institutes of Health1R44CA285136].  Please state what role the funders took in the study.  If the funders had no role, please state: ""The funders had no role in study design, data collection and analysis, decision to publish, or preparation of the manuscript."" If this statement is not correct you must amend it as needed. Please include this amended Role of Funder statement in your cover letter; we will change the online submission form on your behalf. 4. Thank you for stating the following in the Acknowledgments Section of your manuscript: [This work was supported by 1R44CA285136 (Philibert and Dawes, MPI).   This work is dedicated to the memory of Dr. Meg Gerrard of the University of Connecticut whose work to prevent the effects of alcohol and smoking on risk for cancer have had a profound and lasting impact on the investigative team and the field in general. ]We note that you have provided funding information that is not currently declared in your Funding Statement. However, funding information should not appear in the Acknowledgments section or other areas of your manuscript. We will only publish funding information present in the Funding Statement section of the online submission form. Please remove any funding-related text from the manuscript and let us know how you would like to update your Funding Statement. Currently, your Funding Statement reads as follows:  [[National Institutes of Health1R44CA285136].] Please include your amended statements within your cover letter; we will change the online submission form on your behalf. 5. Thank you for stating the following in the Competing Interests section: [Dr. Philibert is the Chief Executive Officer of Behavioral Diagnostics.  The use of cg05575921 to assess smoking status is covered by existing and pending patents including US Patents 8,637,652 and 9,273,358, while the use of DNA methylation to assess alcohol and predict is covered by existing patents and pending patent claims including European Union Patent 3149206.  On behalf of Drs. Philibert, Behavioral Diagnostics and the University of Iowa have a filed an intellectual property claim on the use of DNA methylation to predict AWS and related phenomena].  Please confirm that this does not alter your adherence to all PLOS ONE policies on sharing data and materials, by including the following statement: ""This does not alter our adherence to  PLOS ONE policies on sharing data and materials.” (as detailed online in our guide for authors http://journals.plos.org/plosone/s/competing-interests).  If there are restrictions on sharing of data and/or materials, please state these. Please note that we cannot proceed with consideration of your article until this information has been declared.  Please include your updated Competing Interests statement in your cover letter; we will change the online submission form on your behalf. 6. We note that you have indicated that there are restrictions to data sharing for this study. For studies involving human research participant data or other sensitive data, we encourage authors to share de-identified or anonymized data. However, when data cannot be publicly shared for ethical reasons, we allow authors to make their data sets available upon request. For information on unacceptable data access restrictions, please see http://journals.plos.org/plosone/s/data-availability#loc-unacceptable-data-access-restrictions.  Before we proceed with your manuscript, please address the following prompts: a) If there are ethical or legal restrictions on sharing a de-identified data set, please explain them in detail (e.g., data contain potentially identifying or sensitive patient information, data are owned by a third-party organization, etc.) and who has imposed them (e.g., a Research Ethics Committee or Institutional Review Board, etc.). Please also provide contact information for a data access committee, ethics committee, or other institutional body to which data requests may be sent. b) If there are no restrictions, please upload the minimal anonymized data set necessary to replicate your study findings to a stable, public repository and provide us with the relevant URLs, DOIs, or accession numbers. Please see http://www.bmj.com/content/340/bmj.c181.long for guidelines on how to de-identify and prepare clinical data for publication. For a list of recommended repositories, please see https://journals.plos.org/plosone/s/recommended-repositories. You also have the option of uploading the data as Supporting Information files, but we would recommend depositing data directly to a data repository if possible. Please update your Data Availability statement in the submission form accordingly. 7. Please note that your Data Availability Statement is currently missing [the repository name and/or the DOI/accession number of each dataset OR a direct link to access each database]. If your manuscript is accepted for publication, you will be asked to provide these details on a very short timeline. We therefore suggest that you provide this information now, though we will not hold up the peer review process if you are unable. 8. Please amend either the title on the online submission form (via Edit Submission) or the title in the manuscript so that they are identical.

Reviewers' comments:

Reviewer's Responses to Questions

**Comments to the Author**

1. Is the manuscript technically sound, and do the data support the conclusions?

Reviewer #1: Yes

Reviewer #2: No

2. Has the statistical analysis been performed appropriately and rigorously? 

Reviewer #1: Yes

Reviewer #2: No

3. Have the authors made all data underlying the findings in their manuscript fully available?

Reviewer #1: Yes

Reviewer #2: Yes

4. Is the manuscript presented in an intelligible fashion and written in standard English?

Reviewer #1: Yes

Reviewer #2: Yes

5. Review Comments to the Author

Reviewer #1: Thank you for the opportunity to read this interesting and well written manuscript that explored the association of epigenetic biomarkers of heavy alcohol drinking and smoking, with all-cause mortality. Findings indicated that Alcohol T Score was significantly related to all-cause mortality in smokers who developed and did not develop lung cancer. DNA methylation at cg05575921 did not have significant association with mortality. The author conclude that heavy alcohol consumption is a significant risk factor of all-cause mortality in smokers. The present study has several strengths including advanced modeling, long follow-up time for death, using participants from case-control study. Despite these strengths, there are several areas that limit my enthusiasm for the study. Below describe the major and minor weaknesses:

Key Weaknesses

1. In the results, you mentioned estimated survival split with AST median, but in the methods, you did not clearly state this analysis. Please add more details for it in the method.

2. Alcohol consumption had different burden on women and men due to their difference on body size, muscle mass, body fat and hormone level. Can you add sex-stratified analysis to further investigate the effect of heavy alcohol drinking on all-cause mortality?

3. Figure3 shows that ATS has a strong correlation with cg05575921. Please provide a more detailed explanation of how the adjusted R-squared value and p-value were calculated. Additionally, clarify the method used to generate the red line in the figure. Furthermore, Figure 3 appears to depict a negative relationship between ATS and cg05575921. Please explicitly state the direction of correlation in the text.

4. Can you add a test for interaction between ATS and cg05575921 in the cox model?

Additional Minor Weaknesses

1. In the introduction, line 70-72 “In abstinent individuals, ATS is zero-centered metric with a standard deviation of 2.2 that non-linearly increases as a function of increasing chronic alcohol consumption.” needs more clarification. Does this mean that, among non-drinkers, ATS has a mean of zero and a standard deviation of 2.2? Additionally, does the statement imply that if non-drinkers start drinking, then the relationship between alcohol consumption and ATS becomes non-linear?

2. In the Methods, the format of each small part is not consistent. Can you add subtitle for each part to improve consistency and readability? Suggested subtitles include Study sample, DNA methylation, AST value, Outcome, statistical analysis.

3. In the results, line 151 “LC cases were not more likely than controls to be current smokers (45% vs 39%, NS). ” What does “NS” mean? What was the p value to support your statement “LC cases were not more likely than controls to be current smokers”?

4. In the results, line 167-168 “median of 20.7 years (interquartile range; 5460-8521 days).”, can you convert the unit of days into years for consistency and clarity?

5. The sentence in the discussion line 205-207 “In a study using a marker for recent (past 3 weeks) heavy alcohol consumption, referred to a ZSCAN25, we found a 0.56 correlation” requires clarification. Could you specify which two variables the 0.56 correlation refers to? Additionally, explain how this correlation supports the claim that "binge drinking is not perfectly correlated with the ATS."

6. In the discussion, line 208-211 “Therefore, we believe that our findings with respect to mortality reflect the entirety of lifestyle risk biology, including risks from poor diets and lack of exercise, that track with the heavily tobacco use and chronic alcohol consumption. ” could benefit from further clarification. Specifically, how do the findings support the conclusion that risks from poor diet and lack of exercise are reflected? Are there specific references? Expanding on this would strengthen the argument.

7. In the discussion, line 222-224 “effects of alcohol measured through this method are greater than those that can be shown using cg05575921, and that with greater sample size, the effects assessed by cg05575921 will become significant.” What specific outcome is being affected by alcohol or cg05575921? Providing this information will help the reader understand the context and implications of these effects more precisely.

Reviewer #2: The authors investigated the associations among cg05575921 (a DNA methylation marker for smoking prediction), the alcohol methylation T score (ATS), and all-cause mortality in lung cancer patients and controls. The results indicate that the ATS score is associated with longer survival times. While the manuscript addresses an important topic, however clarification of the methods and presentation of results are needed to enhance its impact and clarity. Specific questions and comments regarding the manuscript are as follows:

1. The study focuses on two epigenetic markers for smoking and alcohol consumption and investigates their associations with mortality. Although methylation markers reflect a person’s cumulative lifestyle exposure, it would be more informative to compare these markers with actual lifestyle variables (e.g., smoking pack-years and alcohol consumption levels). While I understand that one of the major motivations of the study is the known unreliability of self-reported data in epidemiologic studies, including these variables and comparing the results from the epigenetic signatures could strengthen the findings.

2. Abstract: In lines 45–47, the phrase “in this small group” needs clarification.

3. Also, why did the authors control the occurrence of lung cancer in the analysis? Did the model for cg05575921 include both lung cancer patients and controls? If so, this approach appears inconsistent with that used for the ATS; please clarify.

4. The dataset comprises lung cancer patients and cancer-free individuals, with significant associations observed in both groups. However, the manuscript refers to “at risk for LC” and lung cancer screening efforts. Could the authors provide more contextual information on this point? If space in the abstract is limited, consider including additional details in the Methods section.

5. What is the rationale for providing sex-stratified descriptive statistics? It may be helpful to include P-values for comparisons of demographic and clinical characteristics in Table 1. In lines 156–159, the authors note differences in methylation values by sex. If such differences exist, why were the main association analyses not stratified by sex?

6. Line 110: the abbreviation “EEMS” is used. Please provide the full term when the abbreviation first appears.

7. Methods: The association was investigated using Cox proportional hazards models; however, the details of the models are not well specified. For instance, did the regression model include cg05575921 and ATS simultaneously? Clarification of the modeling strategy is needed.

8. Covariates: Was information on cancer stage and grade available for lung cancer patients? If so, these variables should be included in the analyses. Additionally, please explain why race/ethnicity was excluded from the model construction.

9. Figures: The point estimates mentioned in the text are not displayed in Figures 3 and 4. I am also uncertain whether Figures 1-3 provide sufficient additional information to warrant their inclusion in the main text. Reorganizing the presentation of the findings (both tables and figures) in the Results section would help readers follow the key messages more clearly.

10. Lines 187–189: the authors draw a conclusion that I find difficult to support based on the results presented. Specifically, the analyses were not conducted exclusively among smokers, and referring to “baseline alcohol consumption” is misleading when it is measured via a proxy marker (i.e., the methylation signature). Please clarify how these conclusions were derived from the results.

6. PLOS authors have the option to publish the peer review history of their article (what does this mean? ). If published, this will include your full peer review and any attached files.

**Do you want your identity to be public for this peer review?** For information about this choice, including consent withdrawal, please see our Privacy Policy .

Reviewer #1: No

Reviewer #2: No

---

## [Author Response · Author response to Decision Letter 1]

10 Feb 2025

Editors of PLOS One

February 10, 2025

Please find the attached revised manuscript entitled “Epigenetic Assessments of Heavy Alcohol Consumption Predict Mortality in Smokers at Risk for Lung Cancer in the Prostate, Lung, Colorectal and Ovarian Cancer Screening Trial” as a research article. The original submission received two reviews and eight Editorial comments, several of which require a response. Please see the response to the Editorial comments, then Reviewer comments.

Editorial Comments:

Comment: 1. Please ensure that your manuscript meets PLOS ONE's style requirements,…

Response: We will do our best. We are now using the PLOS EndNote library.

Comment: 2… the ‘Funding Information’ and ‘Financial Disclosure’ sections do not match.

Response: We seemingly cannot get access to the Financial Disclosure section to edit it. Would you please put in “This work was conducted using funding from the National Institutes of Health grant 1R44CA285136 award to R.P. and K.D. The website for the National Institutes of Health is https://www.nih.gov/. The sponsors did not play any role in the study design, data collection, analysis, decision to publish or preparation of the manuscript”? Thank you!

Comment: 3… Please state what role the funders took in the study.

Response: As usual for NIH funded R Series studies, the funders had no role in study design, data collection and analysis, decision to publish, or preparation of the manuscript. Please accept this as our statement. As per the above, we cannot get access to the portion of the website that would allow us to edit that state.

Comment: 4… Please remove any funding-related text from the manuscript…

Response: We have done this per the editor’s request

Comment: 5…Competing interests section….. Please confirm that this does not alter your adherence to all PLOS ONE policies on sharing data and materials…

Response: Our amended Competing Interests section should state “Dr. Philibert is the Chief Executive Officer of Behavioral Diagnostics. The use of cg05575921 to assess smoking status is covered by existing and pending patents including US Patents 8,637,652 and 9,273,358, while the use of DNA methylation to assess alcohol and predict is covered by existing patents and pending patent claims including European Union Patent 3149206. On behalf of Drs. Philibert, Behavioral Diagnostics and the University of Iowa have a filed an intellectual property claim on the use of DNA methylation to predict AWS and related phenomena. This does not alter our adherence to PLOS ONE policies on sharing data and materials.” Thank you for changing this for us.

Comment: 6. We note that you indicated that there are restrictions to data sharing for this study…

Response: As per our data sharing agreement with the National Cancer Institute, their data may not be shared with third parties. Per our funding agreement, the data are being deposited with the the Prostate, Lung, Colorectal and Ovarian (PLCO) Cancer Screening Trial repository. To gain access to that or any other PLCO data, you must submit a project proposal as described at https://cdas.cancer.gov/plco/. We have made that clear in the data availability statement.

Comment: 7. Please note that your Data Availability Statement is currently missing the repository name…

Response: We now have included the CDAS website information.

Comment: 8. Please amend either the title on the online submission form…

Response: Done as requested. A “s” was inadvertently added to the title on the website. It was removed.

Reviewer One

Comment: In the results, you mentioned estimated survival split with AST median, but in the methods, you did not clearly state this analysis. Please add more details for it in the method.

Response: This clarification was made as requested by the Reviewer.

Comment: … Can you add sex-stratified analysis to further investigate the effect of heavy alcohol drinking on all-cause mortality?

Response: These models were run and mentioned in the Results per the Reviewer’s request. The results of the stratified models indicate that the associations of ATS and cg05575921 to mortality do not appear to be impacted by sex. That is, the parameter estimates are similar in models that adjust for sex versus models that stratify by sex.

Comment: Please provide a more detailed explanation of how the adjusted R-squared value and p-value were calculated…clarify the method….and explicitly state the direction of the correlation in the text.

Response: Done per the Reviewer’s request.

Comment: Can you add a test for interaction between ATS and cg05575921 in the cox model?

Response: We did, but the effect was not significant, so it was dropped from the model. We have noted this in the text. But as the Reviewer well knows, the failure to demonstrate the significance is simply a matter of power. We have requested funds to conduct the ATS on all 5000 subjects in our collection. If we get that funding, I am almost certain that there will be a significant interaction effect.

Minor Comments

Comment: 1. In the introduction, line 70-72 “In abstinent individuals, ……..

Response: Yes! The Reviewer understands this exactly. Epigenetic responses are inherently non-linear and non-normally distributed. This is why our more advanced commercial models all use AI (which handles the non-linearity better) and why the use of reference-free approaches, such as MSdPCR, for assessing DNA methylation is necessary for the most accurate descriptions of epigenetic responses to environmental variables. We only normalize the “zero point” for the ATS to make it more interpretable. I just wrote a rather lengthy review of the ATS and ZSCAN25 markers (that was recently published in Epigenomics) that went through the pros and cons of this approach, as well as other related issues, in detail. But I doubt that many will read it.

Comment: 2…the format of each small part is not consistent. Can you add subtitle for each part to improve consistency and readability?

Response: Done per the Reviewer’s request.

Comment: 3…. What does “NS” mean? What was the p value to support your statement…

Response: Done per the Reviewer’s request.

Comment: 4… can you convert the unit of days into years for consistency and clarity?

Response: Done per the Reviewer’s request.

Comment: 5….. line 205-207… Could you specify which two variables the 0.56 correlation refers to? Additionally, explain how this correlation supports the claim that "binge drinking is not perfectly correlated with the ATS."

Response: Done per the Reviewer’s request. Specifically, we expanded the text here to make it more clear what we trying to convey. The ATS is good at picking up sustained alcohol intake, but its dynamic response is slow. Fortunately, there are other markers that respond more quickly. But the ATS does a lousy job at spotting self-reported binge drinking in our experience.

Comment: 6. In the discussion, line 208-211….. Specifically, how do the findings support the conclusion that risks from poor diet and lack of exercise are reflected? Are there specific references? Expanding on this would strengthen the argument.

Response: Done per the Reviewer’s request. Unsurprisingly, so many of these factors have significant collinearity. As a result, it is difficult to assign precise causality.

Comment: What specific outcome is being affected by alcohol or cg05575921?

Response: Good point. We have now added text to further explicate our views on this. As the Reviewer knows, the ATS and cg05575921 are simply good, yet imperfect, biomarkers for the behaviors that drive changes in their values. We have made that point clearer.

Reviewer Two

Comment: 1…. Although methylation markers reflect a person’s cumulative lifestyle exposure, it would be more informative to compare these markers with actual lifestyle variables (e.g., smoking pack-years and alcohol consumption levels). While I understand that one of the major motivations of the study is the known unreliability of self-reported data in epidemiologic studies, including these variables and comparing the results from the epigenetic signatures could strengthen the findings.

Response: The Reviewer has a good point that we are in the process of addressing. We do not yet have the self-report data on alcohol and the self-report data (e.g., pack years and current) on smoking does not add to prediction in this small data set. It simply is a matter of power and we do not wish to others to conclude that we are denigrating the value of the packyear measure because our sample is too small to show effects. Pack years matter-but not for this construct with this number of subjects.

We are just finishing our cg05575921 assessments on all 5000 subjects in our population and plan to address the issue of smoking self-report in the greater context of lung cancer risk prediction. I think that the Reviewer will find that article refreshing. But to be clear, in our extensive experience using these markers in a variety of settings, we find that the value of smoking self-report can be highly variable. In the PLCO data set, the degree of reliability with respect to smoking is considerably higher than many-including the FHS. Unfortunately, since the alcohol assay is much more expensive and time consuming to conduct, we will not have ATS data to accompany that paper. But we have submitted a U01 grant to get those funds as per below.

As far as the alcohol self-report. Well, to be blunt, the reliability of the alcohol self-report data in most public data sets is poor and in actual clinical samples, it is very poor (we cover this in several of our cited articles). We are eagerly awaiting the DHQ data that has the alcohol data on the PLCO population from NCI and have submitted a grant proposal to examine its reliability in the PLCO population. We thank the Reviewer for their patience in the matter. But this issue should be dealt with large numbers of subjects in publicly available datasets for all to peruse. And that is what we plan to do once we get funding for the alcohol assays-which are much more expensive to conduct.

Comment: 2. Abstract: In lines 45–47, the phrase “in this small group” needs clarification.

Response: Done per the Reviewer’s request.

Comment: 3. Also, why did the authors control the occurrence of lung cancer in the analysis?

Response: The analyses were conducted separately because of the large effects of Lung CA on mortality and the fact that the sample is a matched case control study (3:1) based on Lung CA status and additionally matched for smoking, ethnicity and sex. Each model included cg05575921 and ATS because each of the values is known to separately predict mortal risk in and above Lung CA status.

Comment: 4. … Could the authors provide more contextual information on this point? If space in the abstract is limited, consider including additional details in the Methods section.

Response: I see that by the below comments that the Reviewer did not understand that all of these subjects were smokers and former smokers. This is our fault for not making this clearer that all of these subjects were smokers or former smokers and that our overarching purpose is create a rationale for the addition of thorough alcohol screening to all lung cancer risk screening visits. We have added repeated mentions that these subjects are smokers or former smokers as well as test to the end of the introduction making the study rationale clearer.

Comment: 5. What is the rationale for providing sex-stratified descriptive statistics?

Response: Smoking and drinking behaviors differ by gender. So, we standardly present them in our publications. But to address the Reviewer’s comment, we now have noted those characteristics in which male and female subjects significantly differ. In brief, the ATS, mortality and survival were different in the male non-LC controls than in the female non-LC controls.

Comment: 6. Line 110: the abbreviation “EEMS” is used. Please provide the full term when the abbreviation first appears.

Response: Done per the Reviewer’s request.

Comment: Methods: The association was investigated using Cox proportional hazards models; however, the details of the models are not well specified… Clarification of the modeling strategy is needed.

Response: This clarification was made as requested by the Reviewer.

Comment: Covariates: Was information on cancer stage and grade available for lung cancer patients? If so, these variables should be included in the analyses. Additionally, please explain why race/ethnicity was excluded from the model construction.

Response: Yes, those data are available. But none of the patients had lung cancer at sampling, so one cannot put them into the prediction model. Furthermore, even if one did, it is difficult to know what “stage’ one would include. Over the course of ~13 years of observation, 92 of them developed lung cancer, that progressed from Stage 1 (which could have already been present in many cases unknowingly) and for the majority of the subjects with lung cancer, to stage 4 before they died or their data was censored. So, it is not possible to put stage into the initial model.

Race and ethnicity were not included in the model their presence was not significant. But that is not remarkable given the limited number of subjects. We have added this point to the results.

Comment: 9. Figures: The point estimates mentioned in the text are not displayed in Figures 3 and 4. I am also uncertain whether Figures 1-3 provide sufficient additional information to warrant their inclusion in the main text. Reorganizing the presentation of the findings (both tables and figures) in the Results section would help readers follow the key messages more clearly.

Response: If the Reviewer would be more explicit as to how it should be reorganized, we would be happy to oblige. However, Reviewer One did not find the presentation of results confusing and we have used this method of presenting results quite literally dozens of times previously. But are open to explicit suggestions.

Comment: 10. Lines 187–189: the authors draw a conclusion that I find difficult to support based on the results presented. Specifically, the analyses were not conducted exclusively among smokers, and referring to “baseline alcohol consumption” is misleading when it is measured via a proxy marker (i.e., the methylation signature). Please clarify how these conclusions were derived from the results.

Response: The Reviewer is in error. All the subject data analyzed in this paper were from smokers or former smokers. The non-smokers in the PLCO sample were all excluded. We apologize for not making that clearer and have made this more clear at several points of the manuscript. Furthermore, all markers, including those of self-report, are proxy measures, when examining complex human behaviors such as a smoking and drinking. But we see how “baseline alcohol consumption” could be misconstrued and have changed it to “average alcohol consumption at the time of study intake significantly predicted of the overall mortality risk of subjects who were current or former smokers…”.

Thank you for your consideration of this revised manuscript. I look forward to seeing the response from the Reviewers. We stand ready to make additional changes if necessary and desirable.

Sincerely yours,

Robert A. Philibert M.D., Ph.D.

Professor of Psychiatry and Biomedical Engineering

Member, Neuroscience and Genetics Programs

---

## [Decision Letter · Decision Letter 1]

28 Mar 2025

Epigenetic Assessments of Alcohol Consumption Predict Mortality in Smokers at Risk for Lung Cancer in the Prostate, Lung, Colorectal and Ovarian Cancer Screening Trial

PONE-D-25-03475R1

Dear Dr. Philibert,

We’re pleased to inform you that your manuscript has been judged scientifically suitable for publication and will be formally accepted for publication once it meets all outstanding technical requirements.

Kind regards,

Chunyu Liu, PhD

Academic Editor

PLOS ONE

Reviewers' comments:

Reviewer's Responses to Questions

**Comments to the Author**

1. If the authors have adequately addressed your comments raised in a previous round of review and you feel that this manuscript is now acceptable for publication, you may indicate that here to bypass the “Comments to the Author” section, enter your conflict of interest statement in the “Confidential to Editor” section, and submit your "Accept" recommendation.

Reviewer #1: All comments have been addressed

Reviewer #2: All comments have been addressed

2. Is the manuscript technically sound, and do the data support the conclusions?

Reviewer #1: Yes

Reviewer #2: Yes

3. Has the statistical analysis been performed appropriately and rigorously? 

Reviewer #1: Yes

Reviewer #2: Yes

4. Have the authors made all data underlying the findings in their manuscript fully available?

Reviewer #1: Yes

Reviewer #2: Yes

5. Is the manuscript presented in an intelligible fashion and written in standard English?

Reviewer #1: Yes

Reviewer #2: Yes

6. Review Comments to the Author

Reviewer #1: I did not have any additional comments. All my previous questions were solved. And the manuscript is well written.

Reviewer #2: (No Response)

7. PLOS authors have the option to publish the peer review history of their article (what does this mean? ). If published, this will include your full peer review and any attached files.

**Do you want your identity to be public for this peer review?** For information about this choice, including consent withdrawal, please see our Privacy Policy .

Reviewer #1: No

Reviewer #2: No

---

## [Editor Report · Acceptance letter]

PONE-D-25-03475R1

PLOS ONE

Dear Dr. Philibert,

I'm pleased to inform you that your manuscript has been deemed suitable for publication in PLOS ONE. Congratulations! Your manuscript is now being handed over to our production team.

Kind regards,

on behalf of

Dr. Chunyu Liu

Academic Editor

PLOS ONE